**Data Availability Statement:** Archived data is available at figshare: Vegetation data: https://doi.org/10.6084/m9.figshare.c.3143716.v1 Fence

# Naïve plant communities and individuals may initially suffer in the face of reintroduced megafauna: An experimental exploration of rewilding from an African savanna rangeland

Truman P. Young [1,2☯] *, Duncan M. Kimuyu[2,3☯], Wilfred O. Odadi[2,4☯], Harry B. M. Wells[5,6☯], Amelia A. Wolf[7☯]

1 Department of Plant Sciences and Ecology Graduate Group, University of California, Davis, Davis, CA, United States of America, 2 Mpala Research Centre, Nanyuki, Kenya, 3 Department of Natural Resources, Karatina University, Karatina, Kenya, 4 Department of Natural Resources, Egerton University, Egerton, Kenya, 5 Sustainability Research Institute, School of Earth and Environment, University of Leeds, Leeds, United Kingdom, 6 Lolldaiga Hills Research Programme, Nanyuki, Kenya, 7 Department of Integrative Biology, University of Texas, Austin, TX, United States of America

☯ These authors contributed equally to this work.
* tpyoung@ucdavis.edu

## Abstract

Excluding large native mammals is an inverse test of rewilding. A 25-year exclosure experiment in an African savanna rangeland offers insight into the potentials and pitfalls of the rewilding endeavor as they relate to the native plant community. A broad theme that has emerged from this research is that entire plant communities, as well as individual plants, adjust to the absence of herbivores in ways that can ill-prepare them for the return of these herbivores. Three lines of evidence suggest that these "naïve" individuals, populations, and communities are likely to initially suffer from herbivore rewilding. First, plots protected from wild herbivores for the past 25 years have developed rich diversity of woody plants that are absent from unfenced plots, and presumably would disappear upon rewilding. Second, individuals of the dominant tree in this system, *Acacia drepanolobium*, greatly reduce their defences in the absence of browsers, and the sudden arrival of these herbivores (in this case, through a temporary fence break), resulted in far greater elephant damage than for their conspecifics in adjacent plots that had been continually exposed to herbivory. Third, the removal of herbivores favoured the most palatable grass species, and a large number of rarer species, which presumably would be at risk from herbivore re-introduction. In summary, the native communities that we observe in defaunated landscapes may be very different from their pre-defaunation states, and we are likely to see some large changes to these plant communities upon rewilding with large herbivores, including potential reductions in plant diversity. Lastly, our experimental manipulation of cattle represents an additional test of the role of livestock in rewilding. Cattle are in many ways ecologically dissimilar to wildlife (in particular their greater densities), but in other ways they may serve as ecological surrogates for wildlife, which could buffer ecosystems from some of the ecological costs of rewilding. More fundamentally, African savannah ecosystems represent a challenge to traditional

break survey data: https://doi.org/10.6084/m9.figshare.13720492.v1.

**Funding:** The KLEE exclosure plots were built and maintained by grants from the James Smithson Fund of the Smithsonian Institution (to A.P. Smith), The National Geographic Society (Grants 4691- 91, 9106-12, and 9986-16), The National Science Foundation (LTREB DEB 97-07477, 03- 16402, 08-16453, 12-56004, 12-56034, and 19-31224) and the African Elephant Program of the U.S. Fish and Wildlife Service (98210-0-G563) (to T.P. Young, C. Riginos, and K.E. Veblen). The funders had no role in study design, data collection and analysis, decision to publish, or preparation of the manuscript.

**Competing interests:** The authors have declared that no competing interests exist.

Western definitions of "wilderness" as ecosystems free of human impacts. We support the suggestion that as we "rewild" our biodiversity landscapes, we redefine "wildness" in the 21st Century to be inclusive of (low impact, and sometimes traditional) human practices that are compatible with the sustainability of native (and re-introduced) biodiversity.

## Introduction

Ecological restoration has been primarily a botanical endeavor emphasizing re-establishing plant communities (the trophic basis of ecosystems). In contrast, conservation biology has emphasized populations of warm-blooded vertebrates (the species most similar to humans). These differences have divided these two biodiversity disciplines [1]. Species re-introductions, however, have long been a component of biodiversity conservation, and can appropriately be considered a restoration activity. Nonetheless, the fields of conservation and restoration remain largely separate in their practitioners, their taxa of interest (and therefore methods), and their levels of organization (population versus community) [1]. Rewilding, with its emphasis in ecosystem processes [2], has the potential to integrate conservation and restoration [3], albeit still with fundamental differences [4].

Rewilding emphasizes the re-establishment of locally extinct large predators and herbivores [2]. This may be accompanied by either the removal of non-native livestock or their incorporation as wildlife surrogates (e.g., [5]). Large (>100kg) native mammals have been lost from many parts of the world at different times in the past, often at the hands of humans, but the savanna ecosystems of eastern and southern Africa can serve as reference communities of relatively intact mega- and meso-fauna (as well as a millennial-scale history of pastoralism; [6]). More emphasis on rewilding has been placed on large carnivores than on large herbivores [2, 3], despite large herbivores having a much stronger experimental record of ecosystem effects. In addition, examinations of the effects of reintroduced large mammals on plant communities are sparse [2, 7], especially in the form of controlled, replicated experiments [8], and are often assumed to be positive [9, 10].

We know from herbivore (and carnivore) introductions to isolated islands that plant (and animal) communities not adapted to these herbivores may suffer tremendous changes, including species loss [11, 12]. We expect that communities defaunated at different time scales would similarly be dominated by plant species less adapted to herbivory by large mammals. We also know that individual plants relax their (inducible) defenses in the absence of herbivores [13]. These two forms of plant naivete may put individual plants and plant communities at risk during refaunation.

What does the world look like with and without its large native herbivores, and with and without livestock? What lessons does this hold for the rewilding endeavor? The Kenya Long-term Exclosure Experiment (KLEE) has been manipulating three guilds of large native and domestic herbivores in an African savanna for the past 25 years. The selective removal of large herbivores from a more intact ecosystem can be thought of as an "inverse test" of rewilding. Here we synthesize some of the more striking results, as they inform the potential and pitfalls of rewilding as a restorative conservation practice. In particular, we suggest that long-defaunated landscapes may be dominated by "naïve" plant individuals and communities (those that are favoured in the absence of herbivory) that are ill-prepared for the reintroduction of historic populations of large wild herbivores.

## Study site and exclosure design

This research was carried out at Mpala Conservancy, located on the Laikipia plateau in central Kenya (0˚17'N, 36˚52'E). The study site is located within *Acacia drepanolobium* wooded grassland at an elevation of 1800m, on heavy clay ('black cotton') soils. Mean annual rainfall 1995–2017 was 600 mm/yr (range 365–1000 mm/yr), which on average falls in a weakly trimodal seasonal pattern, with a distinct dry season December-March.

The understory is dominated by 5–6 species of perennial grasses (~85% of herbaceous cover), with a rich community of >100 additional plant species (see Supplement 1 in [14]). The woody layer is dominated (>90% of cover) by *Acacia drepanolobium*, which is spinescent and also hosts colonies of four mutually exclusive ant species that vary in their defensive behaviours [15].

The region has been under various forms of cattle management for over three thousand years [6, 16–19]. Most recently (last 100 years) the study site has been a commercial ranching operation increasingly tolerant of wildlife. The Mpala Conservancy is managed for both wildlife conservation and livestock production. Cattle are stocked at moderate densities (0.10–0.15 cattle ha$^{-1}$). Wild ungulates commonly found in the study site include: plains zebra (*Equus quagga* Gray), Grant's gazelle (*Gazella [Nanger] granti* Brooke), elephant (*Loxodonta africana* Blumenbach), cape buffalo (*Syncerus caffer* Sparrman), eland (*Taurotragus oryx* Pallas), giraffe (*Giraffa camelopardalis* L.), hartebeest (*Alcelaphus buselaphus* Pallas), oryx (*Oryx gazella beisa* L.), steinbuck (*Raphicerus campestris* Thunberg), bush duiker *(Sylvicapra grimmia* L.)*, Grevy's zebra (*Equus grevyi* Oustalet), and common warthog (*Phacochoerus africanus* Gmelin) [20]. Wildlife densities in Laikipia are the second highest in East Africa, after the Mara-Serengeti ecosystem.

In 1995, we established the Kenya Long-term Exclosure Experiment (KLEE), designed to tease apart the separate and combined effects of cattle and wildlife on each other and on the savanna ecosystem that they share. The KLEE experiment uses a series of semi-permeable barriers to allow access by six different combinations of cattle ('C'), native meso-herbivore ungulates 15–1000 kg ('W': mainly zebras, gazelles, eland, hartebeest, oryx, buffalo; "Wildlife" hereafter) and mega-herbivores ('M': elephants and giraffes). The experiment consists of three replicate blocks separated from one another by 70–200 m. In each block, there are six random-stratified 200 x 200 m (4 ha) treatment plots (18 total plots; 24ha). The six treatments are: 1) MWC–accessible to mega-herbivores, meso-herbivore wildlife and cattle; 2) MW–accessible to mega-herbivores and meso-herbivore wildlife; 3) WC–accessible to meso-herbivore wildlife and cattle; 4) W–accessible to meso-herbivore wildlife; 5) C–accessible to cattle; and 6) O–no large herbivore access.

Herds of 100–120 mature cows (sometimes with calves) are grazed in each cattle-treatment plot for two hours on each of two to three consecutive days, typically 3–4 times per year. These grazing and herding practices replicate typical cattle management on most private and some communal properties in the region. The cattle are in an individual plot for only a few hours per year, greatly reducing the possibility that wildlife responses are due to direct avoidance of cattle. For full details of the basic experimental design, see [21, 22].

## Herbaceous and woody data collection

Herbaceous vegetation data have been collected in all 18 KLEE plots annually (in June) or biannually (in February and June) since 1999. Here, we use aerial plant cover data collected by counting the number of pins hit by each species of grass or forb over a ten-point pin frame at gridded sampling stations spread evenly over the innermost hectare of each plot. Details of the sampling are given in 20. Because our primary interest was species composition, here we use

only relative cover data. Relative cover is the total number of pins hit by each species divided by the total number of pins hit across all species within each plot and sample period. Relative cover provides an index of the contribution of each species to the herbaceous community while controlling for differences in total biomass due to herbivore treatments and rainfall variability. For our assessment of rare species (Table 2), we used frequency data, based on presence/absence from 100 1m x 1m quadrats in each plot. In 2011 (after 16 years of exclosure), we also surveyed all 18 KLEE plots for woody species, enumerating every individual encountered.

On 19 July 2014, there was a break in one of the fences that allowed elephant entry into plots from which they had previously been excluded until the fence was repaired on 31 July. In late 2014 and 2015, we carried out surveys of recent elephant damage to *Acacia drepanolobium* in plots that a) had been exposed to elephants continually, and b) had been protected from elephants and giraffes from 1995 until 2014 but exposed to elephants during to the fence break. We concentrated our surveys on the trees around glades, landscape features that have higher densities of larger trees and are preferred browsing sites for elephants. For each *A. drepanolobium* tree encountered, we measured height, stem diameter, and distance from glade edge, and assessed elephant damage using the following classes: 0: no damage, 1: 1–24% damage 2: 25–49% damage, 3: 50–74% damage, 4: 75–99% damage, 5: 100% damage, complete breakage of above ground stem.

## Statistical analyses

Mean values were calculated for each plot, and one-way and two-way ANOVAs were run on plot means, with the three blocks as replicates.

**Ethics statement.** This research was carried out under Government of Kenya research clearance permit No. 347 NCST/RCD/12B/012/42

## Results

### Woody plant diversity in KLEE plots

We found a total of 38 species of woody plants in the KLEE plots in 2011, after 16 years of exclusion of different large mammalian herbivores. Of these, 15 were found only in plots from which native large herbivores had been excluded (Table 1). Mean woody plant species richness varied from 7.5 to 20, and increased with increasing exclusion of native large mammals (F = 148.1, p < 0.001; Fig 1).

### Elephant damage on naïve and experienced A. drepanolobium trees

We surveyed a total of 550 individuals of *A. drepanolobium*, 202 in three plots to which elephants had continuous access, and 348 in two plots from which elephants had been excluded for 20 years, until the temporary fence break. Estimated elephant damage differed significantly between size classes (larger trees suffering more damage) and past elephant exposure (naïve trees suffering more damage). There was a significant interaction between these two factors (p<0.001); the greater damage in naïve trees was only significant among trees >4m tall (F = 22.78, p = 0.018; Fig 2). Damage was more than twice as high among trees occupied by the ant *Crematogaster sjostedti* than other trees (P<0.001). Elephants also attacked other woody species, including those that had recruited over the previous 20 years, but their rarity or absence in plots exposed to elephants precluded similar comparisons.

**Table 1. List of the 15 woody plant species found only in KLEE plots protected from all large herbivores (treatment O) or plots accessible only to cattle (treatment C) in the 2011 survey, and therefore likely to disappear if large native herbivores are reintroduced (in order of abundance in protected plots).**

| |
| --- |
| *Kleinia squarrosa* |
| *Lantana viburnoides* |
| *Aloe* sp. A |
| *Phyllanthus sepialis* |
| *Plectranthus sp.* |
| Unidentified sp. 1 |
| *Ximenia americana* |
| *Ipomea* sp. |
| *Maerua angolensis* |
| *Asclepias* sp. |
| *Ormocarpum* sp. |
| *Rhamnus staddo* |
| Unidentified sp. 2 |
| *Aloe* sp. B |

## Differences in herbaceous community structures after 25 years of large herbivore treatments

The herbaceous communities after 25 years of KLEE treatments had diverged considerably, especially in the plots protected from both cattle and large wild herbivores (Treatment O in Fig 4, [20]). In particular, plots protected from large mammal herbivory were dominated by *Brachiaria lachnantha*, the grass most preferred by cattle [23] and *Pseudognaphalium* sp., a perennial forb that, although unpalatable, is sensitive to herbivore disturbance (Fig 2 in [20]). More than twenty of the less common plant species were only recorded in (all of) our protected plots, especially those protected from megaherbivores (Table 2).

## Discussion

### Excluding large native mammals as an inverse test of rewilding

If we demonstrate how the exclusion of large herbivores affects an ecosystem, can we "reverse engineer" [24] the effects of adding native large mammals back to the ecosystem? How might a defaunated plant community respond to the reintroduction of large herbivores? How might such a defaunated plant community respond to the replacement of cattle by native wildlife? The KLEE plots provide insights into these questions.

### Naivete as a risk in reintroduction

Naivete to re-introduced consumers can occur at the community, population and individual levels. As we have shown in KLEE, the removal of large mammalian herbivores can shift the herbaceous plant community in the direction of more palatable species and/or species with relatively low tolerance to defoliation. The return of large herbivores may come as a severe "shock" to this savanna community, as these less well-defended plant species are heavily utilized, and eventually replaced by less palatable species, resulting in a major shift in community structure.

There are two kinds of plant naivete that may produce at least transient costs of re-introductions. First, the individual organisms may be ill-prepared for life in the newly re-faunated community and are at risk until they develop more defensive traits. This concern has rarely

**Table 2. Herbaceous plant species found only in surveys of all six plots protected from all native large herbivores (List A), or only in all 12 plots protected from only native mega-herbivores (List B).**

| |
| --- |
| List A: Species observed in all treatments without either wildlife or megaherbivores (O and C), but never in with native large herbivores (W, WC, MW, MWC). |
| *Achyranthus aspera* |
| *Amaranthus hibridus* |
| *Chenopodium opulifolium* |
| *Cyphostemma* sp. |
| *Ocimum sp.* |
| List B: Species observed in all treatments without megaherbivores (O, C, W, WC), but never in treatments with megaherbivores (MW, MWC) |
| *Achyranthus aspera* |
| *Amaranthus hibridus* |
| *Chenopodium opulifolium* |
| *Chenopodium schraderianum* |
| *Conyza pedunculata* |
| *Cyathula orthacantha* |
| *Cynodon dactylon* |
| *Cynodon plectostachyus* |
| *Cyphostemma sp* |
| *Emilia discifolia* |
| *Harpachne schimperi* |
| *Melhania velutina* |
| *Mollugo nudicaulis* |
| *Monsonia angustifolia* |
| *Ocimum sp* |
| *Pavonia patens* |
| *Schkuhria pinnata* |
| *Sida sp.* |
| *Sporobolus sp.* |
| *Tagetes minuta* |
| *Unidentified fern* |
| *Unidentified lily* |

Data from surveys in 2017, 2018, and 2019. For the meaning of the treatment acronyms, see Methods.

been raised for plants, although occasionally for animals (e.g., introducing naïve animals into the wild: [25, 26]).

Second, members of the recipient community may be dominated by species not evolved to receive herbivores or carnivores that have been long absent. An extreme example of this may be the extinctions of naïve megafauna following the arrival of humans in Australia, the Americas, and various island ecosystems [27], but it might also occur with more recent carnivore re-introductions (c.f. [28]).

Two kinds of individual plant defenses react to the presence or absence of consumers, at different time scales. Constitutive defenses are increased or decreased over evolutionary time, whereas induced defenses react on shorter time scales within an individual plant's lifetime [13, 29]. However, both may leave individual plants at risk when suddenly confronted by a long-absent consumer population.

By excluding herbivores, the KLEE experiment has induced naivete in the plant community and provides insight into the cost of plant naivete in the face of herbivore re-introduction.

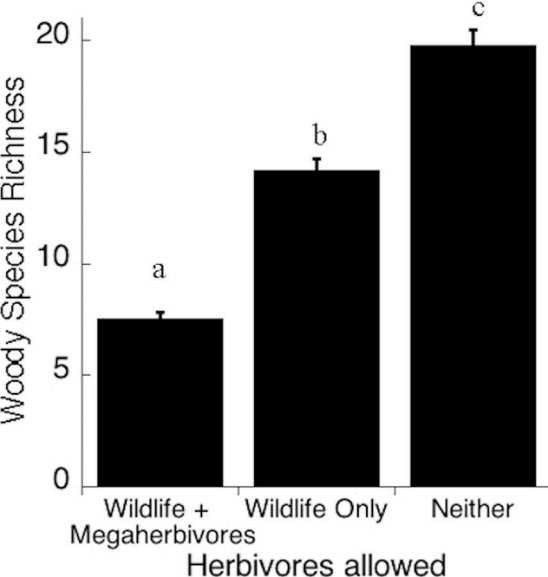

**Fig 1. Woody species richness in the KLEE plots in 2011 (after 16 years of large herbivore exclusion).** Cattle had no significant effect on woody species richness, and each bar includes data from treatments with cattle and without cattle. "Wildlife" here refers to non-megaherbivore large herbivores (see Methods). Bars represent one standard error (three replicates). For the meaning of the treatment acronyms, see Methods.

*Acacia* species are classic examples of plants exhibiting induced defense. In the KLEE plots, spine lengths and chemical defenses decrease in the absence of herbivores [30–32]. In addition, *A. drepanolobium* trees protected from herbivory are significantly more likely to be occupied by *Crematogaster sjostedti*, the least protective of the four ant species [33], likely because

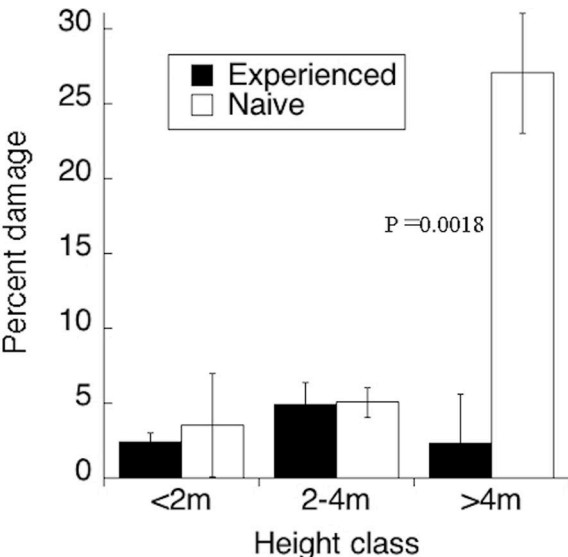

**Fig 2.** Damage by elephants to *Acacia drepanolobium* trees that had either a) been continuously exposed to native herbivores, or b) protected from elephant herbivory for 20 year before a break in the wildlife fence (Naïve). Bars represent one standard error. For the meaning of the treatment acronyms, see Methods. Bars not sharing a letter are significantly different.

protected trees reduce their rewards to strongly defensive ants [32]. Even if defensive recovery is relatively rapid (as little as weeks) after reintroduction of herbivores, there is plenty of time for herbivores to do severe damage to not-yet-well-defended plants.

In addition to our "reverse engineered" tests of herbivore reintroduction from the KLEE project, the 2014 incursion by elephants offers a more direct text of the naivete hypothesis. Indeed, the naïve trees suffered far higher elephant damage than the more "experienced" trees that have been continually exposed to elephants (Fig 2) and have maintained their physical (spines) and symbiotic (ants) defenses [32, 34]. This damage was greatest on trees occupied by the non-defensive ant *C. sjostedti*, which is the ant species most associated with plots protected from herbivory [33].

Any re-introduction of a consumer population to a community from which it had been long absent could result in similar losses. These would be expected to be more severe when the defenses lost were constitutive (evolutionary) and therefore slower to respond. The loss of plant defenses in the KLEE experiment over the past 25 years are almost entirely due to relaxation of induced defenses. The absence of herbivores on millennial time scales would likely also include evolutionary (constitutive) reductions in defenses to large herbivores.

## Community changes in plots protected from large mammalian herbivores

Not only can individual plants become naïve to large mammal herbivory, so can entire plant communities, when the absence of herbivores allows more susceptible species to flourish. In KLEE, protection form large native mammals resulted in large increases in the species richness of the woody layer (Fig 1).

Were these large herbivores to be re-introduced into these plant communities, one would predict the species richness of the woody layer to decline rapidly, including through local extinctions. In fact, we are planning just such a reversal of the experimental treatments, which will provide the opportunity to test this hypothesis. Planned reversals of KLEE treatments will provide even more direct tests of rewilding.

An interesting comparison can be made with the situation of wild (escaped) horses in the United States [35], which are broadly believed to be responsible for various degradative effects in these ecosystems ([36, 37], see also [38]). Yet multiple native equids were part of the Pleistocene megafauna of North American [39], and presumably targets for re-wilding [40]. Their modern "damaging" effects may be attributable to either a) the absence of appropriate Pleistocene carnivores that may have kept their numbers low, or b) precisely the kinds of community responses expected from plant communities that have become naïve to such herbivores over the last 10,000 years. Similar arguments might be made for cattle west of the Great Plains (and their once-massive bison herds). The KLEE experiment has been in place since 1995 (25 years), and therefore evolutionary relaxations of plant defense are unlikely to have occurred.

## Additional rewilding lessons from the KLEE project

A recent increase of elephants in the broader Laikipia ecosystem in which KLEE is embedded provides yet another unplanned test of rewilding, with KLEE acting as an experimental control. Over the past 20 years, elephants have significantly increased in Laikipia [41], perhaps because of the deteriorating conservation status in elsewhere in the region. Patterns from our study site match widely reported impacts of elephants and other large herbivores on woody vegetation in this ecosystem and elsewhere [42–45]. *Acacia drepanolobium* densities have declined over the past 20 years in unprotected KLEE plots only (Table 1 in [46]). Even more pronounced, these plots have also lost most of their *A. mellifera* trees. During the same interval of increasing elephant numbers, populations of two other *Acacia* species in adjacent plant

communities, *A. hockii* and *A. seyal fistula*, have been completely extirpated. For a similar experience of increasing elephant populations in southern Africa, see [47]).

In addition, we have shown experimentally that although elephants alone have only modest effects on *A. drepanolobium* populations, in combination with fire, another natural disturbance in this ecosystem, their effects are great [46]. Conversely, we have shown that herbivory by large mammals can reduce the intensity, severity, and continuity of fires [48, 49]. Together, these results suggest that the ecosystem effects of refaunation will interact with fire and other ecosystem processes in complex ways.

It is worth noting that the plant species excluded from KLEE plots to which large herbivores have access are present in the broader ecosystem, which has resident populations of these herbivores. KLEE is situated in a relatively homogeneous community, both biologically and topographically. A broader, more topographically diverse landscape may harbour local refuges from large mammal herbivory [40, 50, 51].

## Our experimental exclusion of cattle represents one of the secondary goals of rewilding

Cattle are clearly keystone species in this ecosystem. They have myriad impacts that may be considered degradative, and some effects directly resulting from the techniques of cattle management [22, 52, 53]. These effects support the removal of livestock as a potential goal of rewilding. On the other hand, results from KLEE show that cattle (at moderate densities) are in many ways (though not all) ecological surrogates for wildlife (20, see also [54]). Note that the plant species in Table 2 all occur in the presence of cattle, but not native wildlife. Overall herbivore pressure may be more influential than the particular mix of native and domestic herbivore species, and the replacement of livestock with wild herbivores may result in less dramatic effects on plant populations and communities than rewilding into landscapes with no recent history of livestock. A similar case has been made for bison and cattle in North America (e.g., [55, 56]). Worldwide, the presence of domestic herbivores may have buffered ecosystems to some of the ecological costs of large mammal defaunation [54]. Similarly, the reintroduction of carnivores, by reducing wildlife numbers or habitat use [7, 9], may also buffer ecosystems from some of the effect the reintroduction of wild herbivores.

## At moderate densities: Redefining "wildness" in the 21st Century

Does rewilding need to minimize/eliminate the human footprint? In particular, multiple ecosystems in Africa can be test cases of what could work, and what does not. There has recently been a strongly expressed disagreement about the appropriate meanings of "wilderness" and "wildness", often in the context of rewilding [57, 58]. The traditional Western concept of wilderness, as expressed in the 1964 U.S. Wilderness Act and IUCN (Category Ib: Wilderness Area), defined wilderness as areas "untrammeled by man" and "without permanent or significant human habitation", respectively. This definition, rooted in human-nature or culture-nature dualisms, has recently been challenged on historical, geographic, political, social, and conservation grounds [52, 57, 58]. In particular, the tremendous diversity of large mammals in African savannah ecosystems has been (until recently) compatible with millennial-scale pastoral human presence [52]. Over the past two decades the KLEE project, in one such African ecosystem, has experimentally demonstrated that considerable large mammal diversity can be compatible with livestock production, *at moderate densities* [20, 22, 53, 59]. Hunter-gatherer societies are likely to have been even more compatible with the maintenance of biodiversity. This supports a broader definition of wildness that could form a basis of rewilding, in Africa and elsewhere [4, 58].

## Acknowledgments

We would like to thank Frederick Erii, John Lochikuya, Mathew Namoni, Jackson Ekadeli, and Stephen Ekale for their invaluable assistance in the field. We also thank the Mpala Research Centre and its staff for their logistical support. Useful comments came from Johan du Toit, Jennifer Adams Krumins, and Lauren Porensky.

## Author Contributions

**Conceptualization:** Truman P. Young.

**Data curation:** Truman P. Young, Duncan M. Kimuyu.

**Formal analysis:** Truman P. Young, Duncan M. Kimuyu, Harry B. M. Wells.

**Funding acquisition:** Truman P. Young.

**Investigation:** Truman P. Young, Duncan M. Kimuyu, Wilfred O. Odadi, Harry B. M. Wells.

**Methodology:** Truman P. Young.

**Project administration:** Truman P. Young, Duncan M. Kimuyu, Wilfred O. Odadi, Amelia A. Wolf.

**Supervision:** Truman P. Young.

**Writing – original draft:** Truman P. Young.

**Writing – review & editing:** Duncan M. Kimuyu, Wilfred O. Odadi, Harry B. M. Wells, Amelia A. Wolf.

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
