## [Decision Letter · Decision Letter 0]

19 Jan 2021

PONE-D-20-34917

Naïve plant communities and individuals may initially suffer in the face of reintroduced megafauna: an experimental exploration of rewilding from an African savanna rangeland

PLOS ONE

Dear Dr. Young,

Thank you for submitting your manuscript to PLOS ONE. After careful consideration, we feel that it has merit but does not fully meet PLOS ONE’s publication criteria as it currently stands. Therefore, we invite you to submit a revised version of the manuscript that addresses the points raised during the review process. As you will see, both reviewers suggest to include more information about the ponetial limitations (mostly spacial and temporal) of your study. On the other hand, they suggest to incorporate some details that I am sure will improve the quality of your research. 

We look forward to receiving your revised manuscript.

Kind regards,

Emmanuel Serrano, PhD

Academic Editor

PLOS ONE

Journal Requirements:

2.) Thank you for stating the following in the Acknowledgments Section of your manuscript:

'The KLEE exclosure plots were built and maintained by grants from the James Smithson Fund of the Smithsonian Institution (to A.P. Smith), The National Geographic Society (Grants 4691-91, 9106-12, and

9986-16), The National Science Foundation (LTREB DEB 97-07477, 03-16402, 08-16453, 12-

56004, 12-56034, and 19-31224) and the African Elephant Program of the U.S. Fish and Wildlife

Service (98210-0-G563) (to T.P. Young, C. Riginos, and K.E. Veblen). Useful comments came from Johan du Toit and Lauren Porensky.'

'No: The funders had no role in study design, data collection and analysis, decision to publish, or preparation of the manuscript.'

3.) We note that you have stated that you will provide repository information for your data at acceptance. Should your manuscript be accepted for publication, we will hold it until you provide the relevant accession numbers or DOIs necessary to access your data. If you wish to make changes to your Data Availability statement, please describe these changes in your cover letter and we will update your Data Availability statement to reflect the information you provide.

Reviewers' comments:

Reviewer's Responses to Questions

**Comments to the Author**

1. Is the manuscript technically sound, and do the data support the conclusions?

Reviewer #1: Yes

Reviewer #2: Yes

2. Has the statistical analysis been performed appropriately and rigorously? 

Reviewer #1: Yes

Reviewer #2: Yes

3. Have the authors made all data underlying the findings in their manuscript fully available?

Reviewer #1: Yes

Reviewer #2: Yes

4. Is the manuscript presented in an intelligible fashion and written in standard English?

Reviewer #1: Yes

Reviewer #2: Yes

5. Review Comments to the Author

Reviewer #1: I appreciate having the opportunity to review this work, which is based on an account of the consequences of a serendipitous breach of the fence around an exclosure experiment in Kenya. After 20 years of protection from elephant browsing, the larger Acacia drepanolobium trees had apparently relaxed their defenses, with the defensive ant community having shifted to a less aggressive composition. There might also have been changes in spine length and leaf chemistry, though those were not reported. Anyhow, the larger A. drepanolobium trees (>4 m high) had become “naïve” to elephants and seemingly also more palatable, and were damaged much more during the period that elephants had access to them than “experienced” trees that had been continually exposed to elephants. The authors use this case study to illustrate a thoughtful discussion about how rewilding a landscape through introductions of large herbivores might lead to an episode in which certain plant species are impacted much more than expected, because they have become “naïve”. This is a good point, and a helpful contribution to the current discussion on rewilding. I have two overall comments and several specific ones.

Overall:

1). The scale dependence of megaherbivore effects need to be considered. This study is at the scale of 4 ha exclosure plots in a topographically homogenous landscape where the woody vegetation is strongly dominated by one species. At the spatial scale of an elephant home range (100s of square km), in a landscape with more diversity in topography (hills, rivers, plains) and plant community types, the services performed by elephants (nutrient cycling, seed dispersal, differential disturbance, tree felling that provides protection to seedlings beneath the recumbent canopy, etc., etc.) would be more evident. Certainly, as in this case, prolonged protection from large herbivores can be associated with reduced inducible defenses in plants, which are then vulnerable when large herbivores suddenly reappear, but some plant species are continuously vulnerable to large herbivores and persist in a source-sink dynamic system in which refugia are crucial. For example, baobab tree populations can be eradicated by elephants unless there are rocky hills on which some individuals persist and provide propagules that find their way into the areas below (Edkins et al. 2007). Also, riparian tree communities maintain diversity in the face of chronic elephant browsing by virtue of small stands persisting above steep river banks (du Toit et al. 2014). So, if rewilding projects involving large herbivores consider the issue of scale, and if topographic diversity is adequately incorporated into the reserve design, then the problem of plant naiveté should be greatly mitigated.

2). The discussion about the effects of cattle on herbaceous vegetation does not fit with the story about elephants selectively damaging tall A. drepanolobium trees. It is a distraction, especially for readers who do not know the functional traits of the various herbaceous species listed (Table 3, Fig. 3). I suggest keeping the focus on the woody vegetation.

Specifics:

Page 3: In the introduction it is stated that rewilding is often associated with the elimination of non-native livestock, but I disagree. To the contrary, livestock species are commonly promoted as surrogates for extinct or rare wild species and are key players in well-known rewilding projects in Europe, for example, including Oostvaardersplassen in the Netherlands.

Page 3: Spelling of Beschta.

Page 4: When elephants broke into the plots, for how long were they feeding there before the exclosure fence was repaired?

Table 3: This is difficult to interpret and seems tangential to the study.

Figure 1: Are megaherbivores not wildlife? Where is the bar for the plots in which cattle were allowed to feed?

Figure 3: This is difficult to interpret and seems tangential to the study.

References:

du Toit, Skarpe & Moe (2014) Elephants and heterogeneity in savanna landscapes. Pages 289-298 in Skarpe, du Toit & Moe (eds.) Elephants and Savanna Woodland Ecosystems. Zool. Soc. Lond. and Wiley-Blackwell.

Edkins, Kruger, Harris & Midgley (2007). Baobabs and elephants in Kruger. Afr. J. Ecol. 46:119-125.

Signed: Johan du Toit

Reviewer #2: Review

Title: Naïve plant communities and individuals may initially suffer in the face of reintroduced

megafauna: an experimental exploration of rewilding from an African savanna

rangeland

Authors: Truman P. Young, Harry Wells, Duncan Kimuyu, Wilfred Odadi, Amelia Wolf

Overview:

Young et al. present data that emerged from a long-term herbivory exclosure experiment. They show that when plants are protected from the grazing activities of mega herbivores, they lose their defenses, become naïve and more vulnerable to the subsequent return of grazers. This paper is brief, clear and the data supports their conclusions well. The results they find here are important to a large body of fundamental literature that studies the effects of herbivory on plant communities as well as the literature addressing the science of conservation and rewilding. I appreciate their attention to both.

The conclusions of this paper would benefit from an acknowledgment of the variability of time. Diversity of the plant communities is higher in exclosures after 25 years. I am not sure that would be the case if herbivores were excluded in perpetuity. I also feel that future work that includes genetic analysis, particularly on the acacia would be incredibly interesting especially given they can age the trees. Broadly, the paper is very well written. I have a few minor hesitations that will be described below in comments.

Comments:

1. Please always include line numbers to assist your reviewers.

2. I strongly encourage the authors to remove the value laden statements. This paper is submitted as a ‘research paper’ not as an opinion piece or synthesis in which the reader may expect presentation of the authors opinions. As such, I also encourage them to remove the quote at the end of the discussion. The data and conclusions of this work are clear, compelling and speak for themselves. Given this is a research article, please allow your readers to make their own interpretation regarding the value of rewilding and its implications for conservation.

For example, in the abstract: “In summary, the native communities that we observe in defaunated landscapes may be very different from their pre-defaunation states, and we are likely to see some large (and potentially troubling) changes to these plant communities upon rewilding with large herbivores.” Potentially troubling to whom? This sentence otherwise clearly describes your results, but its strength is lost when it becomes a statement of your opinion.

3. Table 1 is more of a list than a comparative table. I suggest moving that to an appendix if possible.

4. Table 2 appears unformatted. It is also not necessary given the statistics presented in the figures. I encourage you to remove it or format it formally.

5. Can you please remind the readers what O, C, MW and MWC designate? Do this for Figure 3 and Table 3.

6. Figure quality appears low. I assume the journal will require higher resolution images. They were tough to read (esp fig 3).

7. Replication should be stated clearly in the methods and presented again in each figure caption. Is it three given six treatments and 18 plots? This is not clear.

reviewed by Jennifer Adams Krumins

6. PLOS authors have the option to publish the peer review history of their article (what does this mean?). If published, this will include your full peer review and any attached files.

Reviewer #1: **Yes: **Johan T. du Toit

Reviewer #2: **Yes: **Jennifer Adams Krumins

---

## [Decision Letter · Decision Letter 1]

3 Mar 2021

PONE-D-20-34917R1

Naïve plant communities and individuals may initially suffer in the face of reintroduced megafauna: an experimental exploration of rewilding from an African savanna rangeland

PLOS ONE

Dear Dr. Young,

Thank you for submitting your manuscript to PLOS ONE. After careful consideration, we feel that it has merit but does not fully meet PLOS ONE’s publication criteria as it currently stands. Therefore, we invite you to submit a revised version of the manuscript that addresses the points raised during the review process mainly related to the quality of figures and the need of including a table.

We look forward to receiving your revised manuscript.

Kind regards,

Emmanuel Serrano, PhD

Academic Editor

PLOS ONE

Journal Requirements:

Reviewers' comments:

Reviewer's Responses to Questions

**Comments to the Author**

1. If the authors have adequately addressed your comments raised in a previous round of review and you feel that this manuscript is now acceptable for publication, you may indicate that here to bypass the “Comments to the Author” section, enter your conflict of interest statement in the “Confidential to Editor” section, and submit your "Accept" recommendation.

Reviewer #1: (No Response)

Reviewer #2: All comments have been addressed

2. Is the manuscript technically sound, and do the data support the conclusions?

Reviewer #1: Yes

Reviewer #2: Yes

3. Has the statistical analysis been performed appropriately and rigorously? 

Reviewer #1: Yes

Reviewer #2: Yes

4. Have the authors made all data underlying the findings in their manuscript fully available?

Reviewer #1: Yes

Reviewer #2: Yes

5. Is the manuscript presented in an intelligible fashion and written in standard English?

Reviewer #1: Yes

Reviewer #2: Yes

6. Review Comments to the Author

Reviewer #1: My main comments from the first review have mostly been adequately addressed and, where not, it is a matter of opinion that the authors are entitled to maintain and the readers to evaluate. I have just a few minor comments:

The first two sentences of the last paragraph of the discussion seem odd. First, the question “Does rewilding need to minimize/eliminate the human footprint?” is moot since du Toit & Pettorelli (2019; referenced) outlined how rewilding is generally inclusive of people with nature (Table 1 of that paper). Second, the sentence “Africa as a test case …” is perhaps intended to read “Africa IS a test case”? Anyhow, extrapolating from the KLEE site to the whole of Africa is quite a stretch and so I suggest some rewording.

Line 375, correct to Ripple & Beschta (not “Breschta”).

Line 437 in the references please correct to du Toit JT (not “JY”).

Reviewer #2: Young et al. have satisfactorily addressed my comments and those of reviewer 1. I am happy to suggest this manuscript be accepted with minor caveats. I still do not understand why the lists are presented in the manuscript itself, and not as a formal table. I still hold they would be far more accessible if presented as a clean table and likely as a supplement. I also encourage the authors to double check the order and quality of their figures. I might be confused, but as they are presented in the revised manuscript, they appear to be ordered figure 2 before figure 1, and quality is still low.

7. PLOS authors have the option to publish the peer review history of their article (what does this mean?). If published, this will include your full peer review and any attached files.

Reviewer #1: **Yes: **Johan T. du Toit

Reviewer #2: No

---

## [Editor Report · Decision Letter 2]

8 Mar 2021

Naïve plant communities and individuals may initially suffer in the face of reintroduced megafauna: an experimental exploration of rewilding from an African savanna rangeland

PONE-D-20-34917R2

Dear Dr. Young,

We’re pleased to inform you that your manuscript has been judged scientifically suitable for publication and will be formally accepted for publication once it meets all outstanding technical requirements.

Kind regards,

Emmanuel Serrano, PhD

Academic Editor

PLOS ONE

Additional Editor Comments (optional):

Congratulations Prof. Truman!
---

## [Editor Report · Acceptance letter]

25 Mar 2021

PONE-D-20-34917R2 

Naïve plant communities and individuals may initially suffer in the face of reintroduced megafauna: an experimental exploration of rewilding from an African savanna rangeland 

Dear Dr. Young:

I'm pleased to inform you that your manuscript has been deemed suitable for publication in PLOS ONE. Congratulations! Your manuscript is now with our production department. 

Kind regards, 

on behalf of

Dr. Emmanuel Serrano 

Academic Editor

PLOS ONE